# Progress in Research on TLR4-Mediated Inflammatory Response Mechanisms in Brain Injury after Subarachnoid Hemorrhage

**DOI:** 10.3390/cells11233781

**Published:** 2022-11-26

**Authors:** Lintao Wang, Guangping Geng, Tao Zhu, Wenwu Chen, Xiaohui Li, Jianjun Gu, Enshe Jiang

**Affiliations:** 1Institute of Nursing and Health, Henan University, Kaifeng 475004, China; 2School of Clinical Medicine, Henan University, Kaifeng 475004, China; 3Department of Neurology, The First Affiliated Hospital of Henan University, Kaifeng 475001, China; 4Henan Technician College of Medicine and Health, Kaifeng 475000, China; 5Department of Geriatrics, Kaifeng Traditional Chinese Medicine Hospital, Kaifeng 475001, China; 6Department of Neurosurgery, Henan Provincial People’s Hospital, Zhengzhou 450003, China; 7Henan International Joint Laboratory for Nuclear Protein Regulation, Henan University, Kaifeng 475004, China

**Keywords:** subarachnoid hemorrhage, TLR4, brain injury, cerebrospinal fluid, inflammatory response

## Abstract

Subarachnoid hemorrhage (SAH) is one of the common clinical neurological emergencies. Its incidence accounts for about 5–9% of cerebral stroke patients. Even surviving patients often suffer from severe adverse prognoses such as hemiplegia, aphasia, cognitive dysfunction and even death. Inflammatory response plays an important role during early nerve injury in SAH. Toll-like receptors (TLRs), pattern recognition receptors, are important components of the body’s innate immune system, and they are usually activated by damage-associated molecular pattern molecules. Studies have shown that with TLR 4 as an essential member of the TLRs family, the inflammatory transduction pathway mediated by it plays a vital role in brain injury after SAH. After SAH occurrence, large amounts of blood enter the subarachnoid space. This can produce massive damage-associated molecular pattern molecules that bind to TLR4, which activates inflammatory response and causes early brain injury, thus resulting in serious adverse prognoses. In this paper, the process in research on TLR4-mediated inflammatory response mechanism in brain injury after SAH was reviewed to provide a new thought for clinical treatment.

## 1. Introduction

Subarachnoid hemorrhage (SAH) is a fatal stroke subtype, with its incidence accounting for about 5–9% of cerebral stroke patients and a mortality of 20–30% [1,2,3,4]. Even surviving patients often suffer from serious adverse prognoses such as hemiplegia, aphasia, cognitive dysfunction and even death, which greatly affects the quality of life of the patients and causes a huge social burden [4,5,6,7]. Until now, increasing evidence has suggested that neuroinflammatory response is one of the key factors for poor prognoses in SAH patients [4,8,9]. However, the exact molecular mechanism underlying the inflammatory response after SAH remains unclear, which hinders the development of effective and specific treatments for SAH.

Toll-like receptors (TLRs) are a family of pattern-recognition receptors that exert a key role in the inflammatory response [10]. In the TLRs family, TLR4 is widely expressed in the central nervous system, including microglia, neurons, astrocytes, endothelial cells, choroid plexus epithelial cells (CPECs), etc. [11,12,13,14]. Recent studies have suggested that the TLR4-mediated signal pathway has an important effect on the pathogenesis of inflammatory response after SAH [5,15]. Additionally, some studies have pointed out that after SAH, the gene expression of many inflammatory mediators, such as tumor necrosis factor-α(TNF-α), interleukin-1 (IL-1) and interleukin-6 (IL-6), can be down-regulated by inhibiting the TLR4 transduction pathway, which reduces the occurrence of nervous system inflammation after SAH and alleviates the brain injury [16,17,18,19]. Moreover, it has been proved that TLR4 plays a vital role in initiating inflammation related to cerebral stroke, Alzheimer’s disease, Huntington’s disease, Parkinson’s disease, etc. [11,18,20,21]. However, the specific mechanism of TLR4 upregulation in SAH patients has not yet been further explored.

Therefore, in this paper, the current progress in research on TLR4 in the neuroinflammatory response after SAH was reviewed. This aims to explore the role of TLR4 in the early neuroinflammatory response in SAH and provide potential therapeutic targets, with the expectation of offering a reference for future clinical treatment.

## 2. Research Progress in Poor Prognosis after SAH

SAH is a cerebral stroke caused by blood flow into the subarachnoid space due to the rupture of blood vessels at the base or surface of the brain, which triggers corresponding clinical symptoms [22]. About 85% of spontaneous SAH cases are caused by ruptured cerebral aneurysms; 10% of cases are caused by non-aneurysmal rupture around the midbrain, and the remaining about 5% of SAH cases are due to various rare congenital or acquired cerebrovascular diseases or systemic ones, such as arteriovenous malformations, moyamoya diseases, cerebral amyloid angiopathy, vasculitis, hemorrhagic diseases, reversible cerebral vasoconstriction syndrome or drug abuse [23,24,25]. SAH is characterized by high fatality and disability rates, with a mortality of 35%, and surviving patients often remain disabled for life [26,27,28]. Thus far, the specific mechanism of brain injury caused by SAH has been unclear.

Cerebral vasospasm (CVS) is one of the devastating complications that occur in SAH patients, and it often leads to delayed cerebral ischemia (DCI), permanent neurological deficits and even death, usually occurring days after SAH [29,30]. Cerebral vasospasm after SAH was initially considered the crucial cause of high mortality and poor prognoses [31]. However, in a later study, it was found that even if the cerebral vasospasm in SAH patients is improved, the mortality and clinical outcome of patients are not significantly improved yet [11,32,33]. This result indicated that cerebral vasospasm might not be the leading cause of the poor outcome of the disease in patients. Therefore, researchers began to pay attention to the new mechanism of nerve injury caused by SAH. Currently, the research on the mechanism of poor clinical prognoses caused by SAH mainly focuses on early brain injury (EBI) and delayed brain injury (DBI) [34,35]. In particular, EBI is considered to be the main reason for the poor outcomes in patients after SAH [34,36,37].

Post-SAH EBI refers to any acute pathophysiological events that occur in the brain within 72 h after SAH, such as brain cell death, blood–brain barrier destruction, cerebral edema, oxidative stress, neurological deficits, acute vasospasm and impaired brain autoregulation [34,38]. In contrast to EBI, DBI refers to the pathophysiological event in brain tissue 72 h after SAH, which is mainly caused by delayed cerebral vasospasm (DCV) [39,40].

In recent years, more and more evidence has suggested that the inflammatory response of the nervous system is throughout in EBI and DBI, especially playing a key role in the pathogenesis of EBI [41,42,43]. Studies have demonstrated that the TLR4- mediated signal pathway exerts an important effect on the inflammatory process of the nervous system after SAH [44,45,46]. Furthermore, in animal experiments, it was found that the inflammatory response after SAH can be reduced by inhibiting the TLR4-mediated signaling pathway, which alleviates EBI after SAH, thus improving the prognosis [4,17,20,32,47,48]. Many studies have pointed out that the up-regulation of TLR4 in the brain after SAH is related to the cognitive dysfunction of prognosis [49]. Therefore, inhibition of the TLR4 signal pathway may be a valuable therapeutic target for EBI after SAH. However, the potential specific mechanism of TLR4- mediated neuroinflammation in SAH remains unclear yet.

## 3. Introduction to TLR4

TLRs, a type I transmembrane protein, are pattern recognition receptors in innate immunity, which were first discovered in the study of Drosophila embryonic development. Additionally, they can be activated by endogenous damage-associated molecular patterns (DAMPs) or exogenous pathogen-associated molecular patterns (PAMPs) ligands, thus initiating relevant signal transduction pathways and playing an important role in immune response and inflammatory response [50,51,52,53]. At present, there are 10 TLR members found in humans. Among them, TLR1/2/4/6/10 are located on the cell surface, and TLR3/7/8/9 are expressed inside the cell [51].

TLR4, which is the first TLRs protein discovered by humans, is an important innate immune receptor [54,55]. Moreover, it is the most widely studied member of the TLR family [54,55]. TLR4 consists of three parts: the extracellular, transmembrane and intracellular domains [56]. Its N-terminal is an ectodomain composed of about 20–30 leucine-rich repeats (LRRs) with a horseshoe-shaped structure, through which PAMPs and DAMPs recognition is mediated [57]. The transmembrane region is made up of a cysteine-rich domain. Its C-terminal intracellular domain contains a highly conserved sequence, which is similar to the IL-1 receptor domain (also known as Toll/IL-1 receptor (TIR) homologous domain) and involved in initiating the downstream signal transduction [58].

## 4. A Typical DAMP: High Mobility Group Box 1 (HMGB1)

After SAH occurrence, different DAMPs are released, and they have the ability to activate immune cells by linking pattern recognition receptors (PRRs) [59]. Among them, HMGB1 is a representative DAMP in SAH.

### 4.1. The Structure of HMGB1

HMGB1 genes, located on chromosome 13q12, are a non-histone protein encoded by a single gene [60]. HMGB1 proteins consist of three unique domains. Among them, the A-box, located at the N-terminal, is highly conservative in evolution; C-tail, located at the carboxyl-terminal, contains 30 repeated aspartic acid and glutamic acid residues and can participate in regulating the binding affinity of HMGB1 to DNA; B-box, located between the above two, is a functional domain that causes an inflammatory response, and it can play the role of pro-inflammatory cytokines to trigger an inflammatory response [61,62]. Both A-box and B-box, which are composed of three α-helices with strong positive charges, constitute the non-specific DNA-binding region of HMGB1. Additionally, they can widely identify and bind particular DNA fragments and play the role of stabilizing nucleosomal structure, regulating DNA repair, recombination and gene transcription, promoting cell differentiation and maturation, etc. [63,64].

### 4.2. Generation and Function of HMGB1 in SAH

HMGB1 is widely expressed in the nuclei of almost all eukaryotic cells in vivo. Under physiological conditions, HMGB1, localized in the nucleus, stabilizes the formation of nucleosomes, regulates gene transcription, etc. [65,66]. On the contrary, under various pathological conditions, HMGB1 can be passively released from necrotic cells or actively secreted from immune cells or non-immune parenchymal cells [66,67]. Extracellular HMGB1 plays the role of DAMPs by binding to TLRs to activate the pro-inflammatory pathway and aggravate inflammatory injury [67].

After the lysis of the erythrocytes in the subarachnoid space, oxyhemoglobin or hemoglobin is released from the erythrocytes and very easily oxidized to methemoglobin [49]. These products can generate ROS, which causes damage to the lipid molecular layer of the cell membrane, thus resulting in damage to such cells as neurons [49]. Studies show that necrosis and apoptosis of nerve cells can occur immediately 10 min after SAH [68]. The research on the SAH rat model by Sun et al. suggested that damaged cortical neurons near the blood clot site passively release HMGB 12 h after SAH, and HMGB1 proteins and mRNA levels are also significantly up-regulated, reaching a peak one day later [65]. Neurons are also the primary source of HMGB1 in the early stage of brain injury, not only in SAH but in cerebral ischemic stroke. HMGB1 released by neurons in the early stage may be an important early upstream factor of inflammatory response after SAH [65]. Furthermore, studies have shown that SAH also triggers HMGB1 release from the smooth muscle of the arterial wall of the diseased vessel. Haruma et al. found that by administering anti-HMGB1 monoclonal antibodies to SAH rats, the translocation and release of HMGB1 in BA smooth muscle cells are significantly inhibited [69]. After SAH occurrence, in addition to the passive release of HMBG1 by neurons and smooth muscle cells of the diseased vascular wall, glial cells, as resident macrophages of the central nervous system, also participate in the active release of HMGB1 [69]. For one thing, extracellular HMGB1 acts on such receptors as TLR4 to induce an inflammatory response, and for another, the released cytokines, in turn, can lead to further release of HMGB1, and form a positive feedback pathway, ultimately producing the “cascade” effect of inflammation [70].

### 4.3. Regulation of HMGB1 in SAH

HMGB1 is elevated within a short time after SAH. The translocation and release of intracellular HMGB1 is a highly regulated process involving multiple mechanisms [71]. The JAK/STAT pathway, a cytokine signal transduction pathway widely existing in the immune system, plays an important role in both neuroimmune inflammatory response and neuronal apoptosis [72]. Similarly, the JAK/STAT pathway is involved in the regulation of HMGB1 expression and nucleus cytoplasmic translocation after SAH [73].

Lysine at positions 28–44 and 179–185 of HMGB1 is called the nuclear localization signal (NLS) region, in which acetylation sites exist [74]. The effective release of HMGB1 requires the acetylation of lysine at its two localization sequence sites. When lysine is deacetylated, HMGB1 tightly binds to chromatin and resides in the nucleus; when acetylation modification occurs, HMGB1 is separated from chromatin and transferred from the nucleus to the cytoplasm [75]. The activation of JAK/STAT pathways is an important upstream signal for promoting the acetylation occurrence of HMGB1 [73,75]. It has been found in studies that the phosphorylation level of the JAK/STAT pathway increased significantly after SAH occurrence [73]. Moreover, HMGB1 induces the expression of multiple inflammatory cytokines such as IL-1, IL-6 and TNF-α after SAH. These inflammatory cytokines further activate JAK/STAT signal transduction, and in turn, JAK/STAT promotes the expression of HMGB1, thus forming a vicious circle and aggravating brain injury [72]. An et al. [72] found that by inhibiting the activation of JAK/STAT, the expression of HMGB1 in the nucleus and its translocation to the cytoplasm decrease significantly after SAH and that the expression of inflammatory factors is reduced markedly.

### 4.4. Biphasic Action of HMGB1

As a typical DAMP, HMGB1 has traditionally been widely recognized for its pro-inflammatory effect after SAH. However, recent studies have shown another aspect of HMGB1: its important role in promoting tissue repair and reconstruction. It has been found in many studies that the HMGB1 signal pathway can promote the repair function in the diseases such as palate trauma in the oral cavity, lung epithelial injury and acute myocardial infarction [76,77,78]. Additionally, the studies of cerebral hemorrhage also confirmed that HMGB1 promotes angiogenesis and nerve regeneration in the late stage of cerebral hemorrhage [79,80,81].

Similarly, HMGB1 is involved in promoting brain tissue remodeling and neurovascular repair after SAH in the late stage of SAH injury. This biphasic action of HMGB1 is based on different redox modifications of cysteine residues [82]. Tian et al. found by injecting HMGB1 under two different redox states into SAH rats that they show different prognoses 7–14 days after SAH. Recombinant HMGB1 can promote the stimulatory activity of cytokines and aggravate brain injury. However, oxidized HMGB1, which fails to stimulate the production of cytokines, facilitates brain recovery by promoting the expression of neurotrophic proteins [82]. This redox modification can occur on the three cysteine residues of HMGB1, namely C23, C45 and C106 [83]. The cysteines of HMGB1 in the quiescent cells are all in a reduced state. When HMGB1 is released outside cells, disulfide bonds are generated between Cys23 and Cys45, but Cys106 remains in a reduced state. Under this state, HMGB1 can bind to TLR4 to promote inflammatory production [83]. The redox modification of the cysteine residue at position 106 of HMGB1 can change its ability to stimulate cytokines. In Tian’s study, two kinds of HMGB1 are used, one kind of HMGB1 at C106 was oxidized, and another at C106 was in a reduced state. When the reduced form of HMGB1 with cytokine-inducing ability is used, it exacerbates brain injury. However, the use of HMGB1 under the oxidized state without the potential of cytokine stimulation can cause a decrease in the production of inflammatory factors and promote brain recovery by up-regulating neurotrophic factors [82]. Therefore, the conversion aimed at the redox state of cysteines at position 106 of HMGB1 after SAH is a therapeutic target with extremely great potential.

### 4.5. Potential Therapeutic Drugs: Glycyrrhizic Acid (GA)

GA, a natural inhibitor of HMGB1 extracted from traditional Chinese medicine, with a small molecular mass, is easy to pass through the blood–brain barrier [84]. Studies have suggested that the application of GA in the SAH rat model can inhibit the inflammatory response induced by HMGB1, reduce the production of inflammatory factors and alleviate the brain injury caused by SAH in rats [67,85]. NMR and fluorescence studies confirmed that GA could bind to the Box-A and Box-B regions of HMGB1, effectively inhibit HMGB1 translocation and reduce HMGB1 levels outside the nucleus, thus down-regulating the level of downstream inflammatory factors [86,87].

In addition to GA, there are also some small molecule drugs that can inhibit HMGB1 secretion, such as ethyl pyruvate, Vitexin, sodium butyrate and1-stearoyl-2-hydroxy-sn-glycero-3-phosphocholine [64,88,89,90,91,92,93,94]. However, these drugs can only inhibit the active secretion of HMGB1 but fail to block the pro-inflammatory activity of HMGB1 that has been passively released into the extracellular space. There are also some steroid anti-inflammatory drugs, such as dexamethasone. Non-steroidal anti-inflammatory drugs, including aspirin and ibuprofen, cannot significantly inhibit the release of HMGB1 into the extracellular space [95].

Most importantly, the binding force of GA at the site where HMGB1 binds to DNA is relatively weak, and GA has little effect on the normal physiological function of HMGB1 [86]. Therefore, GA can possibly become one of the potentially effective drugs to block the inflammatory response after SAH.

## 5. The TLR4-Mediated Signal Transduction Pathway

For SAH patients, TLR4 is usually activated by endogenous DAMPs released from the blood. After SAH occurrence, a large amount of blood enters the subarachnoid space. The produced heme, oxygenated hemoglobin, methemoglobin, peroxidase-2, high mobility group box-1(HMGB1), matricellular proteins, heat shock protein, fibrinogen, etc., bind to TLR4 receptors in the form of DAMPs, and they are all the intracellular components of ruptured cells or gene products generated after SAH [96,97,98,99,100].

During TLR4 signal pathway transduction, firstly, blood-derived DAMPs bind to the ectodomain of the horseshoe-shaped extracellular region of TLR4 with the assistance of the extracellular binding partners’ myoid differentiation factor-2 (MD-2) and the cluster of differentiation 14 (CD14) to induce signal transduction [101]. Then, according to the intracellular domain TIR, TLR4 binds to specific adaptor proteins with TIR structure in the form of TIR-TIR, thereby transmitting extracellular signals to downstream pathways [58,102]. What is unique about TLR4 is that it can induce the inflammatory response through both myeloid differentiation factor 88 (MyD88)-dependent and TRIF signal pathways [5,103].

In addition, it was found in some studies that experimental SAH induced biphasic changes in TLR4 and NF-κB expression in EBI, including an initial peak phase (within 2–6 h) and a continuous rise phase (within 12–48 h). Moreover, some studies have also shown that the MyD88-dependent pathway initiates “early” activation of NF-κB, whereas the TLR4 signaling pathway through TRIF occurs in the “late” activation of NF-κB [4,5,41]. The coordination of “early” and “late” signals is a function unique to TLR4.

### 5.1. The MyD88-Dependent Pathway

MyD88, an important player in intracellular signal transduction, is composed of the N-terminal death domain (DD) and the C-terminal TIR one. Its C-terminal TIR domain can bind to TLR4, and its N-terminal death domain can bind to IRAK family protein molecules and form a signal transduction complex [104].

During the MyD88-dependent TLR4 signal pathway transduction, when TLR4 is activated by DAMPs ligand, the intracellular TIR domain undergoes a conformational change, and the adaptor proteins MyD88 and TIRAP with TIR domain are recruited and bind to the TIR domain of TLR4. Subsequently, MyD88 also recruits IL-1 r-associated kinase 4 (IRAK-4) and IL-1 r-associated kinase 1 (IRAK-1) [105]. Activated IRAK-4 phosphorylates IRAK-1, followed by binding of phosphorylated IRAK-1 to TNF receptor-associated factor 6 (TRAF6) [104]. TRAF6 binds to UBC13 and UEV1A and forms a TRAF6-Ubc13-Uev1A heteromultimeric complex, which catalyzes lysine (Lys) at the 63 site of TRAF6 and forms a polyubiquitin chain, thus enabling the ubiquitination of the target protein [106].

On the other hand, the covalent binding of TAB1 to the amino terminus of TAK1 promotes the phosphorylation of two threonine residues (Thr-184 and Thr-187) and one serine residue (Ser-192) of TAK1. Then, TAB2 interacts with the k63 polyubiquitin chain of TRAF6, which connects TRAF6 to TAK1 and forms a TRAF6-TAK1-TAB1-TAB2 complex. Next, the TRAF6-Ubc13-Uev1A complex catalyzes the transfer of its own k63-linked polyubiquitin chain to Lys 158 (k158) of TAK1, which in turn enables TAK1 to undergo a polyubiquitination reaction [107]. After TAK1 polyubiquitination, TAB2 binds to the polyubiquitin chain of TAK1 through the highly conserved Np14 zinc finger (NZF) domain, which enables the conformation of TAK1 to change, thereby enabling TAK1 activation to exert kinase activity and activate the downstream transduction pathway [108].

TAK1, also known as MAP3k7, is encoded by the Map3k7 gene. It is a member of the mitogen-activated protein kinase (MAPK) kinase kinase (MAPKKK) family and is involved in the transduction of multiple signal transduction pathways. Moreover, TAK1 is a signaling component in the nuclear factor (NF)-κB (NF-κB) and MAPK signal pathways [107,109,110,111]. Functionally, TAK1, as a common upstream key signal molecule of NF-κB and MAPK, is the node controlling these two pathways, with a central regulatory role in the innate immune signal pathway.

#### 5.1.1. The NF-κB Pathway

IκBα, an inhibitor protein of NF-kB, can bind to and inhibit NF-kB and keep it in the cytoplasm in an inactive state, thus preventing NF-kB activation and entry into the nucleus [112]. The N-terminal of IκBα, a signal–response region, contains serine phosphorylation and ubiquitination sites and plays an important role in the degradation of IκBα. In the NF-κB pathway, the TRAF6-TAK1-TAB1-TAB2 complex activates IκB kinases (IKKs), which subsequently phosphorylate IκBα [113]. After phosphorylation, IκBα can be recognized by the ubiquitin ligase complex for ubiquitination, thereby degrading IκBα. The inhibitory state of NF-κB bound by IκBα is released, which activates NF-κB, thus promoting the synthesis and release of inflammatory cytokines.

#### 5.1.2. The MAPK Pathway

After activation, TAK1 can phosphorylate and activate MAP kinase kinase 6 (MKK6), followed by the activation of the MAPK family JNK, p38, ERK and ERK5, and finally, induce AP-1 activation, thus promoting the synthesis and release of inflammatory cytokines [114,115] (Figure 1).

### 5.2. The MyD88 Independent Pathway (theTRIF Pathway)

Both MyD88 and TRIF pathways can trigger NF-κB and MAPK activation. Some studies pointed out that MyD88 and TRIF mediate the early and late activation of NF-κB and MAPK, respectively [116]. In a study in which TLR4 knockout mice were used to construct a SAH model, neuronal apoptosis in the dentate gyrus presented a TLR4/MyD88 signaling pathway-dependent and microglia-dependent property in the early stage and mainly relied on TLR4/TRIF pathway in the late stage [116]. Unlike MyD88, TRIF also has the ability to induce interferon response elements, thereby producing anti-apoptotic interferons [117]. This anti-apoptotic effect of TRIF ensures that the inflammation caused by NF-kB activation can last for a long time.

During TRIF-dependent signal pathway transduction, TRIF connects with TLR4 through TRAM as a bridge after TLR4 activation [118,119]. In addition to the TIR domain, TBMs also include two domains that can bind to TRAF and RIP, respectively, thereby activating downstream signal pathways [120,121,122].

#### 5.2.1. TRIF-Mediated NF-κB and MAPK Activation

TRAF6 and RIP1 can mediate the activation process of NF-κB and MAPKs in the TRIF pathway, respectively. For one thing, after binding to the TRIF domain, TRAF6 mediates the activation of downstream NF-κB and MAPKs [123]. For another, after binding to the TRIF domain, RIP1 binds to TNF receptor-associated death domain protein (TRADD) through its homotypic DD domain [124,125]. As an adaptor protein that acts as a scaffold, TRADD binds to E3 ubiquitin ligase, which enables RIP1 to undergo polyubiquitination modification [125]. Ubiquitinated RIP1 can activate TAK1, thereby further activating NF-κB and MAPKs pathways [123].

After SAH, TLR4-mediated MyD88 and TRIF pathways can both activate the expression of NF-κB and MAPK pathways. The overactivation of NF-κB and MAPK pathways is observed in SAH-induced aseptic inflammation. NF-κB is the most important transcriptional regulator of inflammation-related genes. The activation of NF-κB induces the production of a large number of inflammatory factors, thus eliciting neuronal injury. Similarly, JNK, ERK and p38 undergo phosphorylation in MAPK pathways, indicating that MAPK pathways participate in the inflammatory response after SAH. These can all cause apoptosis of neurons and endothelial cells, expression of inflammatory cytokines, as well as neuronal injury [12]. The reduction in downstream inflammatory factors and the reduction in neural dysfunction were significantly observed by inhibiting the activation of TLR4 pathways in the SAH rat model in many studies [2,19,20,28].

#### 5.2.2. TRIF-Mediated IRF3 Activation

As a key transcription factor for regulating type I IFN expression [126], IRF3 can modulate type I IFN secretion at the transcriptional level. IRF3 exists in the cytoplasm in an inactive monomeric conformation in resting cells. Once the cell is activated, IRF3 immediately undergoes phosphorylation and forms a homodimer IRF3-IRF3 or a heterodimer IRF3-IRF7 with IRF7, which enters the nucleus and causes specific gene expression [127]. In the TLR4/TRIF signal transduction pathway, TRAF3 can bind to the TRIF domain. TRAF3 activity is activated by self-oligomerization, followed by ubiquitination of Lys 63 of TRAF3 and activation of TBK1 and IKKε [128]. TBK1 and IKKε with phosphatase activity activate IRF monomers to dimerize them, followed by the transfer of IRFs from the cytoplasm to the nucleus and finally promote type I IFN expression, thereby inducing INF-β expression [127,129] (Figure 1).

## 6. TLR4 Expression in Different Cells in the Central Nervous System

In the central nervous system, TLR4 is mainly expressed in neurons, microglia, astrocytes, CPECs, vascular endothelial cells, etc. [11,12,13,14]. After SAH occurrence, the expression and activation of these cells can mediate the responses with their own characteristics. Wang et al. found through immunohistochemical analysis of TLR4 after SAH that TLR4 in neurons after SAH is significantly up-regulated [18]. Islam et al. concluded through the analysis of TLR4-specific gene knockout in microglia, astrocytes and neurons in the SAH mouse model that TLR4 is expressed in microglia, and the activation of TLR4 in microglia after SAH can cause nerve injury [130,131]. Thus far, TLR4 up-regulation in neurons and microglia after SAH has been proven. Astrocytes are the most abundant glial cells in the central nervous system, and they play an important role in maintaining the integrity of the blood–brain barrier and supporting neuronal activity [132]. At present, it is still controversial whether astrocytes are activated after SAH, but the activation of their TLR4 has been found in many inflammations of the central nervous system [133,134]. In addition, SAH patients are often complicated with acute hydrocephalus. From the pathophysiological origin of acute hydrocephalus after SAH, excessive production of cerebrospinal fluid by CPECs is one of the physiological mechanisms. There is a growing view that the degree of inflammatory response in the subarachnoid space after SAH determines the influencing degree of excessive cerebrospinal fluid secretion on acute hydrocephalus [135]. Moreover, significant expression and activation of TLR4 in CPECs were observed in a mouse model of ventricular hemorrhage [136]. Therefore, it was speculated by us that the activated TLR4 in CPECs is also involved in the production of hydrocephalus after SAH.

### 6.1. Microglia

Microglia, as resident macrophages of the central nervous system, are essential components of innate and adaptive immune responses [134,137]. Under physiological conditions, microglia usually present a branched morphology, with relatively small cell bodies and comparatively long and abundant branches [138]. At this time, microglia exert the role of immune surveillance in brain tissue and also play an important role in helping neuronal synapse formation and neurotrophy in development [139]. However, once the brain homeostasis is disrupted, such as SAH or ischemic stroke, the cell morphology of microglia rapidly changes, and cell bodies are enlarged, with axons shortened, thick and big, which are amoeba-like [138]. This type of microglia can express and release a series of factors, such as IL-1, IL-6, iNOS, TNF-α, NO and MMP-9, which cause a string of severe consequences, such as cerebral vasospasm, microthrombosis, blood–brain barrier disruption and nerve cell apoptosis. This seriously affects metastasis and progression of the disease in patients [140,141,142].

It was previously thought that after SAH occurrence, peripheral monocyte–macrophages enter the subarachnoid space with the help of adhesion molecules expressed by vascular endothelial cells and recruit to the injured site, causing an inflammatory response. However, more and more studies have shown that the inflammatory cells recruited in the inflammatory sites mainly come from the microglia in the brain.

Microglia play a key role in the regulation of neuroinflammation in the pathogenesis of EBI [143,144]. Microglia activation forms include M1 and M2 types, which have cytotoxic and cytoprotective effects on nerve cells, respectively [145,146,147]. After SAH, TLR4- mediated activation of microglia toward the M1 type is an important pathway for generating an inflammatory response. After SAH occurrence, a large number of DAMPs rapidly activate microglia through TLR4 to transform them into M1 type, initiate various intracellular signal transduction, and trigger the immune response. Some argue that the dichotomy by which microglia are simply divided between M1 and M2 phenotypes is oversimplified [148]. However, this classification provides a bridge to understanding the function of microglia in inflammation. Thus, it is still the most commonly used definition for exploring the role of microglia [149,150].

M1 microglia activation can also cause a cascade, which enables continuous expansion of the inflammatory response. On the one hand, after activation, M1 microglia can cause cell necrosis by releasing defensins and elastases and also release heat shock proteins and extracellular matrix, further activating the TLR4 pathway [134]. On the other hand, inflammatory cells are activated successively after SAH; after activation, Th cells differentiate toward Th1 cells under the action of environmental factors such as cytokines and antigen-presenting cells, and Th1 cells also mainly produce IFN-γ to activate M1 cells [151,152,153].

A large number of pro-inflammatory cytokines and oxidative metabolites produced by M1-type microglia through TLR4/MyD88 and TLR4/TIRF signal pathways after SAH can lead to neuronal apoptosis and tissue injury. Pro-inflammatory cytokines can also interfere with vascular relaxation and contraction, and recruit peripheral immune cells by up-regulating cell adhesion molecules, thereby aggravating brain injury. The activation of ERK, p38 and JNK in the TLR4-related MAPK transduction pathway after SAH can result in increased release of cytokines such as MMPs, proteases and IFNs, which degrades the tight connexin in the blood–brain barrier, thereby causing the blood–brain barrier destruction, brain edema and leukocyte infiltration [45,99,154].

In inflammatory response, M1- and M2-type microglia often coexist [155]. The M1 type accounts for a larger proportion only in the early stage of inflammation. In the late stage, the selective activation of some receptors in microglia can promote the transformation of the M1 type to the M2 type and the repair of injury. For example, the release of anti-inflammatory factors such as IL-10 and neurotrophin inhibits nerve injury and promotes nerve regeneration through phagocytic cell debris [156]. Among them, TLR4 not only plays a role in brain injury but protects brain tissue through anti-inflammatory effects. Moreover, it can induce IFN-β production and play an anti-inflammatory role through TLR4/TRIF/TRAF3 signal pathways [157]. Therefore, according to the bidirectional characteristics of microglia, their therapeutic targets after SAH mainly include two categories: one is to inhibit the pro-inflammatory effect of microglia, and the other is to promote their anti-inflammatory effect.

### 6.2. Astrocytes

Astrocytes are the most abundant glial cells in the central nervous system, with many protrusions. These protrusions, which stretch and fill between the cell bodies of nerve cells and their protrusions, play the role of supporting and separating nerve cells and participate in the blood–brain barrier formation [132,158]. Astrocytes can differentiate into diverse phenotypes under different stimuli, namely, pro-inflammatory type (type A1) and anti-inflammatory one (type A2) [159,160,161]. SAH contributes to the transformation of astrocytes into type A1. The activation of type A1 astrocytes can lead to the destruction of the blood–brain barrier integrity and neuronal injury [162,163].

There is controversy about whether the activation of astrocyte type A1 is directly elicited by TLR4 activation on its surface. Some studies have pointed out that the TLR expression in astrocytes of healthy individuals is relatively low. However, TLR4 is abundantly present on the surface of astrocyte membranes at the time of injury or inflammatory occurrence [134]. Various DAMPs are released into the subarachnoid space after SAH, which promotes the activation of the A1 phenotype, probably through TLR4. In some studies, it was found through the activation of TLR4 by lipopolysaccharide (LPS) that TLR4 induces the production of type A1 astrocytes and releases inflammatory factors and cytotoxic ones, thus leading to neuronal injury [164]. Additionally, some studies have also suggested that the activation of A1 astrocytes is induced by activated microglia through the secretion of pro-inflammatory cytokines [165]. There were experiments in which astrocytes were strictly purified, and the response of astrocytes to TLR4 agonists was measured to determine their ability to respond autonomously to TLR4 in the absence of microglia. The results indicated that the response to TLR4 agonists completely depends on the presence of functional microglia [132]. However, in any case, EBI caused by astrocyte activation after SAH is directly or indirectly related to TLR4-induced inflammation.

### 6.3. CPECs

Acute hydrocephalus is a type of EBI after SAH, and it can cause increased intracranial pressure, enlarged ventricles and structural brain injury, which can lead to cerebral hernia and death in serious cases [166].

After SAH, microthrombus and its blood decomposition products can, on the one hand, acutely block cerebrospinal fluid circulation channels such as the cerebral aqueducts and, on the other hand, obstruct the arachnoid villi, which damages the reabsorption of cerebrospinal fluid, thus causing acute hydrocephalus [136,167]. In addition to the disturbance of cerebrospinal fluid circulation and absorption, acute hydrocephalus can also be elicited by the high secretion of cerebrospinal fluid caused by the inflammation of CPECs after SAH [136,168].

CPECs, a type of glial cells in the central nervous system, are epithelial cells with the most active secretion in the human body. Under physiological conditions, they produce about 400–500 mL cerebrospinal fluid every day and participate in the blood–brain barrier formation [169]. In experiments, an intraventricular hemorrhage (IVH) model was established by injecting autologous blood into the rat brain ventricles, and subsequently, the rat developed hydrocephalus. During this process, TLR4 was observed to be significantly activated [136]. Additionally, the degree of hydrocephalus in the experimental group was found to be markedly reduced by injecting TLR4 antagonist into the rat model of intraventricular hemorrhage [166]. All these suggested that TLR4 is also highly expressed in CPECs and involved in hydrocephalus formation after intracerebral hemorrhage.

After TLR4 activation in CPECs, inflammatory cytokines such as TNFα, IL-1 and IL-6 can be produced through NF-kB, thereby activating STE20/SPS1-related proline/alanine-rich kinase (SPAK) in CPECs. SPAK is the most important regulatory factor of Na^+^/K^+^/2Cl-ion co-transporter (NKCC1) [13]. Next, SPAK binds to, phosphorylates and stimulates NKCC1 on the CPECs apical membrane [170,171,172]. NKCC1 can regulate water and ion transport channels on CPECs to secrete cerebrospinal fluid [173,174,175,176]. In addition to the direct TLR4 activation on CPECs, the production of pro-inflammatory cytokines by microglia, etc., after SAH can also promote the high secretion of CPECs [169]. The drugs targeting TLR4 or SPAK are a potential therapy in the future.

## 7. Conclusions

In summary, brain injury after SAH is a complex pathological process. This review mainly introduces TLR4-mediated neuroinflammation after SAH, but this does not mean that this mechanism is sufficient to explain all SAH-induced brain damage. Brain injury after SAH is not caused by any single mechanism but by a combination of multiple mechanisms. The immune and inflammatory responses play a crucial role. In particular, TLR4- mediated signal transduction pathways exert an important effect on brain injury. After SAH, TLR4 in the central nervous system is activated by endogenous DAMPs, and massive pro-inflammatory cytokines in the body are released through TLR4-mediated MyD88 and TRIF pathways. Additionally, after SAH, different cells in the central nervous system also produce different damage effects after overactivation. In animal experiments, the targeted intervention in TLR4-mediated signal pathway targets has a certain inhibitory effect on brain injury after SAH. These research results have not been translated into clinical practice so far. However, with the continuous deepening of the research on TLR4 after SAH, the distribution of TLR4 in the central nervous system, its corresponding endogenous ligands, key downstream molecules, the action mechanism of its involvement in brain injury after SAH through mediating inflammatory response, etc., have been continuously elucidated. This will bring breakthrough progress in the research and development of drugs targeting TLR4 and its downstream molecules and clinical treatment in the future. Moreover, a new direction and method will be provided for the treatment of SAH.

## Figures and Tables

**Figure 1 cells-11-03781-f001:**
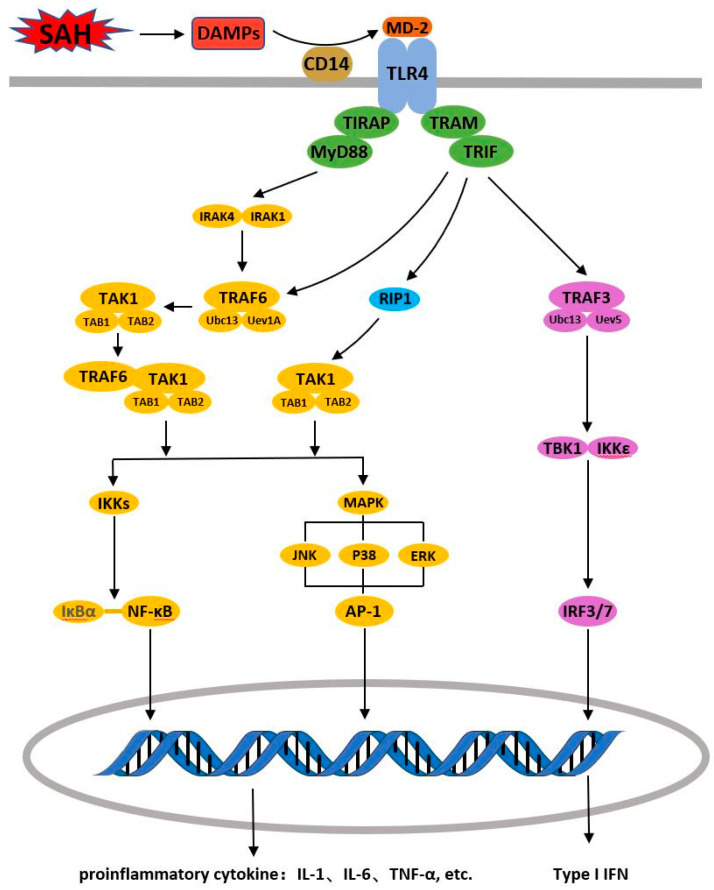
The MyD88 dependent and independent Pathway.

## Data Availability

Not applicable.

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
