# Peer review of "Progress in Research on TLR4-Mediated Inflammatory Response Mechanisms in Brain Injury after Subarachnoid Hemorrhage"

_cells, 2022, doi:10.3390/cells11233781_

Round 1

Reviewer 1 Report (Previous Reviewer 2)

This review compiles recent evidence on subarachnoid hemorrhage (SAH) and the Toll-like receptor 4 (TLR4) that mediate inflammatory response in different type of cells in the brain. SAH is a cerebral stroke subtype caused by blood flow into the subarachnoid space with a 35% mortality. The specific mechanisms of brain damage caused by SAH is still unknown. Thus, identifying the signaling pathways and molecular mechanisms that drive cell death in the brain after SAH is critically important to develop new therapeutic strategies.

This review describes the signaling pathways mediated by TLR4 and discuss the studies on different brain cell types in which TLR4 is expressed and mediates neuroinflammation in SAH models.

The manuscript is well written and organized, and compiles the most recent studies in the field. With the last version of the manuscript, authors have improved the content of the review. There is no comments to address.

Reviewer 2 Report (Previous Reviewer 1)

My concerns have been addressed.

This manuscript is a resubmission of an earlier submission. The following is a list of the peer review reports and author responses from that submission.

Round 1

Reviewer 1 Report

1.   In the manuscript presented by Wang et al., the authors summarized the role of TLR4-mediated inflammatory response mechanism in brain injury after SAH. They mainly reviewed various TLR-mediated signal transduction pathways and TLR4 expression in different cells in the CNS. However, TLR4 is a well-known factor which plays an important role in the pathophysiology of SAH. This manuscript lacks novelty.

2.   The TLR4-medicated signaling pathways introduced in this manuscript are very common and no new pathways were discussed. In addition, the association between these pathways and the pathological process was seldom mentioned.

3.   The expression of TLR4 in different cells should be discussed more in SAH other than CNS.

4.   Though downstream pathways of TLR4 have been reviewed, but they are not enough to illustrate the mechanisms underlying TLR-4-mediated inflammatory response in SAH. The upstream regulation of TLR4 should also be discussed.

Reviewer 2 Report

This review compiles recent evidence on subarachnoid hemorrhage (SAH) and the Toll-like receptor 4 (TLR4) that mediate inflammatory response in different type of cells in the brain. SAH is a cerebral stroke subtype caused by blood flow into the subarachnoid space with a 35% mortality. The specific mechanisms of brain damage caused by SAH is still unknown. Thus, identifying the signaling pathways and molecular mechanisms that drive cell death in the brain after SAH is critically important to develop new therapeutic strategies.

This review describes the signaling pathways mediated by TLR4 and discuss the studies on different brain cell types in which TLR4 is expressed and mediates neuroinflammation in SAH models.

The manuscript is well written and organized, and compiles the most recent studies in the field.

There are only two minor comments:

1.       Authors may consider to include the TRAF6-TAK1-TAB1-TAB2 complex in the Fig. 1.

2.       Is there any evidence on the role of TLR4 in endothelial cells.